# Entrepreneurship in Dairy Cattle Sector: Key Features of Successful Administration and Management

**Georgia Koutouzidou** [1], **Athanasios Ragkos** [2,*], **Alexandros Theodoridis** [3] and **Georgios Arsenos** [4]

1   Hellenic Agricultural Organization-DIMITRA, Institute of Plant Breeding and Genetic Resources, 57001 Thermi Thessaloniki, Greece
2   Agricultural Economics Research Institute, Hellenic Agricultural Organization-DIMITRA, 11528 Athens, Greece
3   Laboratory of Livestock Production Economics, School of Veterinary Medicine, Aristotle University of Thessaloniki, 54124 Thessaloniki, Greece
4   Laboratory of Animal Husbandry, School of Veterinary Medicine, Aristotle University of Thessaloniki, 54124 Thessaloniki, Greece
*   Correspondence: ragkos@elgo.gr; Tel.: +30-210-2755086 (ext. 8)

**Abstract:** In this study, data envelopment analysis is applied to 47 dairy cattle farms to estimate their level of efficiency in the utilization of the available resources and to identify the most efficient ones. The analysis is based on technical and economic data collected through a farm management survey. The main structural and financial characteristics of the most efficient farms are presented, revealing the features that make them better than their peers. A comparative financial analysis is applied between the efficient and inefficient farms, highlighting the appropriate farm structure and determining the major cost drivers in modern dairy cattle farming. The results show that there is still room for improvement in intensive dairy farming. Dairy cattle farms must operate with increased variable cost and utilize their infrastructure at full capacity to decrease their fixed cost per cow. Farms should increase their economic resilience and be less risk averse in an environment where margins to lower production costs in highly intensive farms have been narrowed down. The findings of this study verify that dairy cattle farms of entrepreneurial mindset have the potential to rise to the future economic, environmental and social challenges that will affect the survival of the sector.

**Keywords:** efficiency; dairy cattle farms; management; financial analysis; production cost; feeding patterns

## 1. Introduction

Nowadays, the European dairy livestock sector is characterized by the predominance of intensive production systems that depend on capital endowments, animals of high yields, purchased feedstuff and hired skilled labor [1,2]. The prevalence of such intensive production systems is more evident in the dairy cow sector, where large-scale farms of entrepreneurial type have emerged, driven by the growing demand for milk, the abolishment of milk quotas and high fixed costs [3–5]. The modern European dairy cattle sector utilizes one-third of EU agricultural land, produces massive amounts of milk and rears high-yielding breeds such as Holstein Friesians, which increases its reliance on concentrate feeds [6,7]. The adoption of new technologies in nutrition, genetics and herd management is constantly accelerating, and cows are mainly confined in modern facilities with limited or no access to pastures [2,8].

However, this intensification of the dairy cattle production system leads to hefty expenses [9], animal health and welfare issues (indicatively low fertility, short productive life, mastitis, lameness) and largely negative environmental issues related mainly to GHG emissions, nutrient contamination of soil and groundwater pollution [10]. Hence, a dairy farmer today who operates in a highly competitive but also volatile economic environment

must have specialized knowledge about crop cultivation, animal breeding and nutrition, operation and maintenance of high technology machines and equipment, marketing and financial and human resource management [9,11]. Running a modern dairy farm requires special administrative skills and responsibilities, as the people and the machines that are engaged in the production process call for coordination, training and organization in space and time. The development and the combination of such skills and qualities encumber the management of farms and make them vulnerable to policy changes and to market, technical and environmental challenges [12]. The margins are narrow, and management plays a crucial role in the survival of dairy farms.

Under these circumstances, the economic performance of the modern dairy cattle sector is heavily dependent on efficient management [13], which also largely defines its resilience and sustainability. In the ongoing rapid transition of the sector towards a highly intensive system with explicit entrepreneurial elements [14–16], the efficient use of the existing resources in dairy farms is also crucial to their competitiveness [17,18].

Currently, there is an ongoing debate regarding the solutions that will help meet the expected efficiency in the dairy sector and overcome recurring sustainability challenges [4,12]. In this context, the measurement of efficiency can be a useful tool for revealing the key conditions that are essential for a successful and sustainable business model in European dairy farming [19]. But what are the main features of efficient and profitable dairy farms? The answer to this question is given in this study through an empirical application that identifies the most efficient farms, describes their main structural and financial characteristics and reveals the salient features that make them better than their peers. More specifically, in this study, the efficiency level of 47 modern dairy farms in Greece is estimated through the implementation of data envelopment analysis (DEA) on primary technical and economic data. The dairy farms are stratified into efficiency groups (efficient and inefficient farms) by using the estimated level of efficiency as a classification criterion, and their main technical and financial indicators are calculated and compared, providing an indicative picture of their structure and their productivity.

Recent studies on the evaluation of the managerial abilities of European dairy farmers are mainly focused on the estimation of the technical efficiency (TE) level of the farms and its determinants, without providing insights concerning the operational characteristics and the applied production and management practices of the best farms. Indicatively, Cuesta [20] and Alvarez et al. [17] estimated the efficiency of Spanish dairy farms using stochastic frontier models, Madau et al. [21] evaluated the technical efficiency and the total factor productivity change of dairy farms in EU countries, Kroupová et al. [22] estimated the productivity of Czech farms, while Náglová and Rudinskaya [18] estimated the TE of EU dairy farms and found its determinants. Regarding Greece, the most recent studies on the efficiency of dairy farms were conducted by Theodoridis and Ragkos [23], Siafakas et al. [24] and Mitsopoulos et al. [25]. A review of published studies on the efficiency of dairy farms is also presented in Bravo-Ureta [26]. The applied approaches and the findings of relevant studies are discussed in more detail in Section 3.

We contribute to the existing literature by empirically analyzing primary technical and economic data from modern dairy farms, which follow the structure and the production practices that are commonly applied by most European specialized dairy farms. Thus, the findings and the implications of this study will be pertinent to most European countries. We believe that the outcomes of this study contribute to the debate on finding practical solutions that will enhance the sustainability of the sector and will be used in the future for managerial suggestions and policy recommendations towards a resilient and profitable dairy sector. The limitation of this study is that socio-demographic data were not available to apply a second-stage DEA regression in order to investigate the determinants of inefficiency and that DEA is a non-parametric technique that has no asymptotic properties.

## 2. Materials and Methods

### 2.1. Survey Data

Cow milk production in Greece constitutes an important economic activity, accounting for approximately 20% of the gross value of livestock production in the country. According to the Milk Observatory Board [27], during 2015–2020, milk production was increased by 8% (from 602.9 to 650.6 thousand tons), while dairy farms were reduced by 25%, from 3253 to 2448 farms. This adjustment indicates that there is a clear trend towards the concentration of livestock in a small number of large-sized farms. The technical and economic data for this empirical application were collected in 2017 through a farm management survey in a sample of 47 dairy farms in Greece (Macedonia, Thrace and Thessaly). This area constitutes the main producing center of cow milk in Greece. All sample farms are members of the Holstein Association of Greece (the official breeding body of pure-bred Holstein cows) and constitute 1.5% of Greek dairy cattle farms and 9% of national cow raw milk production. The Holstein Association of Greece has 85 members in total, hence 57.6% were included in the survey. The random sampling technique was applied for the selection of the sample farms All farms deliver their milk to large industries. A questionnaire was designed to record the following data: fixed capital endowments (facilities and machinery, terrestrial improvements, herd size and herd composition); labor requirements and wages (family and hired); land and inputs for the production of feedstuff (acreage and land rent, expenses for seeds, fertilizers, pesticides, fuel, irrigation); purchased feedstuff (quantities and prices); expenses related to animal production (fuel, detergents, electricity, water, drugs, veterinary services, etc.); milk yields and prices; meat yields and prices; value of livestock capital; farm subsidies and compensations. Based on these accounting data, technical and economic indicators, as well as the financial results, were calculated for each farm, providing an indicative picture of their structure and economic performance.

### 2.2. Data Envelopment Analysis Model

An output-oriented DEA model was constructed using the technical and economic data from 47 dairy cow farms, to identify the efficient farms that fully utilize the existing technology of production. DEA is a non-parametric approach that applies mathematical programming techniques to define an efficiency frontier (the limit of the objective capabilities of the production technology) and estimate the level of TE of decision-making units (DMUs), in our case dairy cattle farms, relative to the frontier [28–30]. Each DMU consumes varying amounts of different inputs to produce different outputs, and the level of efficiency is measured relative to the highest observed performance. All deviations from the efficiency frontier are attributed to inefficiencies, while the DMUs that lie on the efficiency frontier are considered fully efficient. The main advantage of DEA is that it does not require specification of the functional form of the production function, compared to stochastic frontier analysis, and it is easy to apply. Moreover, DEA estimates rely on individual observations in contrast to population averages and focus on revealed best-practice frontiers rather than on central-tendency properties or frontier, and it generates the set of "peer" units with which a unit is compared [31].

Assuming that there are n DMUs, each producing a single output by using m different inputs and the $DMU_o$, which represents one of the n DMUs under evaluation, produces $y_o$ units of output using $x_{io}$ units of the ith inputs, the variable returns to scale (VRS) single output-oriented model for $DMU_o$ is expressed as follows:

$$\text{Max } \phi \tag{1}$$

subject to

$$\sum_{j=1}^{n} \lambda_j x_{ij} = x_{io} \quad\quad i = 1, 2, \ldots, m$$

$$\sum_{j=1}^{n} \lambda_j y_j = \phi y_o$$

$$\lambda_j \geq 0 \quad\quad\quad\quad j = 1, 2, \ldots, n \quad\quad\quad (2)$$

$$\sum_{j=1}^{n} \lambda_j = 1$$

i = 1, ..., m inputs; j = 1, ..., n DMUs; where $1 - (1/\phi)$ is the proportional increase in output possible for the ith DMU and λ represents non-negative scalars, expressing the intensity with which a particular activity is employed in production. The output-oriented measure of technical efficiency of a DMU, denoted by TE, can be estimated by

$$TE = \frac{1}{\phi} \quad\quad\quad (3)$$

If $\phi = 1$, then the DMU under evaluation is a frontier point, and it is considered fully efficient, i.e., there are no other DMUs that are operating more efficiently than this DMU. Otherwise, if $\phi > 1$, then the DMU under evaluation is relatively inefficient [29].

DEA is now a well-established method with many applications in the livestock sector, and hence, it is not necessary to go into much detail about its theoretical background. Comprehensive reviews and extensions of the various DEA models can be found in Kumbhakar and Lovell [31], Coelli [32], Cooper et al. [29] and Coelli et al. [33].

The inputs used in the output-oriented DEA model estimated in this study were: (i) farm size (number of dairy cows), (ii) human (family and hired) labor (hours per year), (iii) variable cost (EUR per year) and (iv) fixed capital cost (EUR per year). The input variables were selected to describe the production technology and to reflect the major factors of production utilized in the dairy farms. The output variable was gross revenue (EUR per year). Gross revenue is the value of all outputs produced by the farm in one year (value of milk, value of meat and value of live animals sold) and subsidies and compensations. The pricing system for raw milk in the dairy cattle sector is associated with milk fat and protein content. Higher milk fat and protein content and, hence, higher producer prices are achieved through better management and proper nutrition. Gross revenue, which is a function of prices as well as quantity, was selected as the output measure in the estimation of efficiency in order to take into account the effect of price variability, and consequently the managerial ability of the farmer to produce higher milk fat and protein content, in the output measure. Many studies on the dairy cattle sector use the same input and/or output variables in their efficiency models. Indicatively, gross revenue has been used as output by Náglová and Rudinskaya [18], Theodoridis et al. [19], Theodoridis and Ragkos [23], Mitsopoulos et al. [25], and Spička and Smutka [34]; human labor has been used as input in Alvarez and Arias [17], Náglová and Rudinskaya [18], Madau et al. [21], Theodoridis and Ragkos [23], Siafakas et al. [24], Mitsopoulos et al. [25] and Areal [35]; while variable capital cost and fixed capital cost have been used as inputs by Alvarez and Arias [17], Náglová and Rudinskaya [18], Madau et al. [21],. Theodoridis and Ragkos [23], Mitsopoulos et al. [25], Areal [35] and Kovács and Szücs [36]. Numerous efficiency studies on agricultural and livestock sectors exist that use the same or similar input/output variables.

The output-oriented DEA model was selected for this empirical investigation because (i) dairy cattle farming in Europe and in Greece is rapidly growing in terms of milk production, (ii) consumer's demand for cow milk is increasing, (iii) Greece is deficient in cow's milk, and (iv) the farmers-managers have control over the supply of inputs, and hence, it is reasonable to explore how efficiency could increase the output. The model was estimated using DEAFrontier software [37].

## 3. Results and Discussion

### 3.1. Efficiency Analysis

The frequency distribution of TE estimates obtained from the application of the DEA model is presented in Table 1. The results indicate significant inefficiencies in the performance of the dairy farms. Indeed, under the DEA model, 16 of the 47 farms, i.e., 34.04% of the sample, are fully technically efficient, and the mean TE score is 0.870, indicating that there is substantial inefficiency in farming operations for the sampled dairy farms and suggesting that a 13% increase of the production value is possible, given the level of inputs, provided that the farmers optimize the management of their farms, applying best practices. Only 5 farms, accounting for 10.64% of the sample, exhibit a TE score of less than 70%. The TE score of 8 farms (17.02% of the whole sample) is between 70% and 80%, 11 farms exhibit a score between 80% and 90%, while 7 farms operate close to the efficient frontier with TE between 90% and 100%.

**Table 1.** Frequency distribution of technical efficiency (TE) estimates.

| Efficiency Score | Data Envelopment Analysis | | |
|---|---|---|---|
| | **No. of Farms** | **% of Farms** | **Mean TE** |
| <0.7 | 5 | 10.64 | 0.625 (0.034) |
| 0.7–0.8 | 8 | 17.02 | 0.739 (0.030) |
| 0.8–0.9 | 11 | 23.41 | 0.842 (0.026) |
| 0.9–1.0 | 7 | 14.89 | 0.943 (0.027) |
| 1.0 | 16 | 34.04 | 1.000 (0.000) |
| **Total** | **47** | **100.0** | **0.870 (0.130)** |

Numbers in parentheses are standard deviations.

One of the most appealing characteristics of the DEA approach is that it estimates relative efficiency measures, which are based on the observable sets of "best-practice units", rather than a theoretical maximum [32,33]. Hence, the relatively high mean efficiency score verifies the fact that, nowadays, intensive dairy cow farms operate under a standardized production system that has homogeneous characteristics and does not vary considerably. Another factor that explains the higher TE score is the abolition of milk quotas in 2015. Areal et al. [35] and Čechura et al. [38] found a positive impact of milk quota abolishment on TE, mainly attributed to the scale effect, which was boosted after the abolition of milk quotas and positively affected productivity growth.

Few studies have estimated the level of technical efficiency of dairy cows in Greece using DEA. Theodoridis and Ragkos [23], based on primary data from 2004, reported a TE score of 0.748, while Mitsopoulos et al. [25], who applied an input-oriented DEA model on 116 farms in Greece, found an average score of 0.754. Psychoudakis and Dimitriadou [39] reported a TE score of 0.91, a finding similar to ours, while Siafakas et al. [24], who explored the efficiency of 78 dairy cow farms in Greece in relation to feed resources, reported an average TE score of 0.676. The differences in the efficiency score are partially explained by the different model specifications, as the estimated level of efficiency is affected by the selection of the inputs and the model orientation.

A concise presentation of published studies that estimate the efficiency level of dairy farms can be found in Bravo-Ureta et al. [26]. The authors reported that the mean level of technical efficiency for the non-parametric approaches such as DEA was 78.8%, and for the studies that used cross-sectional data, like ours, the mean TE was 75.5%, a result slightly different from ours. However, since then, the dairy cow sector has undergone a major transition that affected its technical and economic characteristics; therefore, farm heterogeneity (i.e., different production technologies) has been reduced. These changes are expected to be reflected in a higher level of efficiency in the operation of the farms. Indeed, Musliu et al. [40] estimated the technical efficiency of 57 farms in Kosovo using SFA and reported a high TE score of 0.95, a result that is in accordance with our findings. In addition, Náglová and Rudinskaya [18], based on panel data obtained from the Farm Accountancy

Data Network (FADN) database covering EU member states (no data available for Cyprus and Greece) for 2004–2019, calculated an average TE score of 0.900 for EU-27 members. Náglová and Rudinskaya [18] used the stochastic frontier approach (SFA) for the estimation of efficiency, an approach that does not attribute any deviation from the efficient frontier entirely to inefficiencies [30]. A similar analysis was conducted by Kroupová et al. [22], who used the panel data set from the FADN database that covers the period 2004–2016 and 27 EU member states. Using SFA, they calculated an average TE score of 94.01% during the analyzed period, a result that does not align with ours.

The output-oriented DEA method identifies efficiency targets, which are achievable increases in output, while using the same level of inputs [41]. The results of efficiency improvement projection are presented in Table 2. The efficient target of the sampled farms (DEA projection of the optimal output value, which in our application is gross revenue) is EUR 655,196, which represents the gross revenue that would be achieved if farms operated on the efficient frontier, exhibiting a 100% TE score. In this case, the gross revenue would increase by 14.7% (from EUR 571,422 to EUR 655,196). This could be achieved if farms efficiently utilized the existing production technology, holding their inputs constant, indicating that their profitability and consequently their sustainability could be improved substantially, since a higher income would cover the high production cost.

**Table 2.** Average existing and efficiency frontier.

| Improvement Projection | Gross Revenue in EUR |
|---|---|
| Existing output | 571,422 |
| Efficient target | 655,196 |

*3.2. Comparative Analysis*

The estimated TE score was used as a classification criterion and the sample was categorized into two efficiency groups that include the inefficient and the efficient dairy farms, respectively. Moreover, the inefficient farms were further divided into two sub-groups of inefficient farms to get more insight into their characteristics: the first inefficient sub-group includes the inefficient farms that exhibit a TE score from 0.80 to 0.99, while the second sub-group includes the more inefficient farms that exhibit a TE score below 0.80. The main technical indicators of these average efficient and inefficient farms are presented in Table 3, giving a good insight into the structure, organization and productivity of the dairy farms. In the efficiency literature, most studies on livestock indicate a positive relation between herd size and efficiency [23,42–44]. In our study, the two average farms have the same size of approximately 143 cows, verifying the fact that modern dairy farms are large and try to take advantage of scale to reduce the high fixed cost. The size of the farms does not differentiate even among the two inefficient sub-groups (149.8 and 130 cows, respectively). However, efficient farms exhibit higher variation in herd size (143.6 ± 109.6 cows) than inefficient farms (142.2 ± 62.2). Additionally, the cows of the average efficient farm achieve higher milk yields, producing 8.4 tons of raw milk per cow annually, 9.6% higher than the inefficient farms (7.7 tons). The inefficient farms, which, however, have a large TE score (TE from 80 to 99%), achieve high milk yields no larger than the efficient farms. Milk yield is considered a critical factor for the productive performance of a dairy cow farm and one of the main growth drivers of the sector. Considering the low opportunity costs of land and labor, the intensive dairy farms in the EU are expected to further increase milk production in the short term [45]. The increase in milk yield is driven by economies of scale since the cost of production per unit of milk decreases with increasing variable capital [5,46].

**Table 3.** Technical indicators of the technical efficient and inefficient dairy farms.

| Technical Indicators | Efficient Farm (n = 16) | Inefficient Farm (n = 31) | Sub-Groups of Inefficient Farms | |
| --- | --- | --- | --- | --- |
| | | | TE: 0.80–1.00 (n = 18) | TE < 0.80 (n = 13) |
| Cows (number) | 143.6 | 142.2 | 149.8 | 130 |
| Milk yield (kg/cow/year) | 8413 | 7675 | 8324 | 6635 |
| Land (irrigated equivalent in ha) | 0.275 | 0.247 | 0.226 | 0.270 |
| *Non-irrigated (ha/cow)* | *0.015* | *0.037* | *0.091* | *0.098* |
| *Irrigated (ha/cow)* | *0.260* | *0.210* | *0.189* | *0.240* |
| Labor (hours/cow/year) | 81.9 | 97.3 | 98.9 | 94.8 |
| *Family (hours/cow/year)* | *38.0* | *51.2* | *48.1* | *51.9* |
| *Hired (hours/cow/year)* | *43.9* | *46.1* | *50.7* | *42.9* |

Regarding the on-farm production of feed (mainly lucerne, maize and cereals), the efficient farms cultivate on average 0.028 more hectares per cow than the inefficient ones, indicating that disconnection of feed procurement from the markets fosters efficiency. This finding converges with the findings by Alvarez and Arias [17] and Theodoridis and Ragkos [23], who have shown that efficiency is positively related to land size in dairy cow farms. However, according to Siafakas et al. [24] and Mitsopoulos et al. [25], a higher efficiency level was associated with less acreage available for cultivation of feed. Dairy cattle farming is characterized by high land requirements for the on-farm production of feed [47]. In general, during the shifting of dairy cattle farming towards a highly intensive business model over the last two decades, farmers prefer to procure concentrated feed from the market and cultivate crops for forage and silage on-farm. This feeding strategy has resulted in the reduction of the cultivated land to produce concentrated feed and can be mainly attributed to the decoupling of farm subsidies [23,48].

Feeding cost is considered a major cost driver in dairy farming and an important factor of productivity, since it is highly associated with milk yields [45,46]. Therefore, the adoption of an effective feeding strategy is crucial to the economic performance and the sustainability of the farms [47]. To get more insight into this issue, the synthesis of the cultivated land is also provided in Table 3. The results show that the efficient farms use their land mainly to produce forage and silage (0.260 ha/cow compared to 0.210 ha/cow for the inefficient farms) rather than concentrates (0.015 ha/cow for the efficient and 0.037 ha/cow for the inefficient farms), mitigating the uncertainty that prevails in the market of forage crops. In general, specialization in feedstuff produced on-farm increases the control that farms have on feed quality and availability, although they are thus burdened with land rent and costs for machinery and crop storage [48].

The EU dairy cattle sector consists of two main farming types: intensive and mountain farming. The latter accounts for 10% of the milk production in the EU and tends to be small-scale and extensive, contributing to the sustainable development of marginal areas by maintaining landscapes and biodiversity [4]. The former procures feeds exclusively from the markets or produces a small part of the feeds on-farm by cultivating its own or rented land [46]. Recently, in the literature, there is a debate regarding the financial impact of the choice between purchasing or producing feed. Siafakas et al. [24] showed that using land for home-grown feed does not necessarily reduce the cost of feeding; however, the farms that do not cultivate crops for feed and are more focused on milk production tend to be more efficient. This finding coincides with Clay et al. [47], who indicated that very intensive farms that rely only on purchased feed (they do not cultivate land) achieve higher profitability.

The results also show human labor used for animal treatment and feed production is 15.8% less in the inefficient farms (97.3 h/cow for the inefficient and 81.9 h/cow for the efficient). Similar are the results when comparing the efficient farms to both inefficient sub-groups. This is in line with Theodoridis and Ragkos [23], Siafakas et al. [24], Mitsopoulos et al. [25] and Sauer and Latacz-Lohmann [46]. The efficient farms tend to organize labor more

effectively, while they implement labor-saving innovative technologies that include remote monitoring (cow collars, high-tech pedometers), robot milking and cleaning systems and automated feeding systems [48]. The synthesis of labor requirements indicates that the share of hired labor is higher in the efficient farms (43.9 h/cow, i.e., 53.6% of total labor, while in the inefficient it is 47.3%), verifying the entrepreneurial nature of these dairy farms that hire specialized workers with specific skills in dairy farming.

In general, the composition of gross revenue does not differentiate much between the two major efficiency groups ($p$ = 0.629) (Table 4). The results confirm that milk production is the predominant activity, since the lion's share of gross revenue comes from milk, contributing by 87% in both efficiency groups. The average milk price for the milk delivered to industries by the efficient farms was only 1 cent higher (0.435 EUR/lt in the efficient farms compared to 0.425 EUR/lt in the inefficient). The contribution of meat and breeding for selling animals to other farmers in gross revenue is trivial and verifies that income depends heavily on the fluctuations of the farm-gate milk prices as well as on milk yields. Indeed, the sale of calves for breeding and beef meat accounts just for 1.6% in both major efficiency groups, while the share of the value of the cows sold for breeding and of the veal meat in gross revenue is higher in the efficient farms (6.7% compared to 5.9% in the inefficient farms). This income dependence on one product (milk) reduces farmers' resilience to income shocks [49]. In addition, vulnerability is exacerbated by the share of subsidies in gross revenue, which is higher in the inefficient farms, although at low levels for both efficiency groups (5.6% and 4.4% for the inefficient and efficient farms, respectively). The share of subsidies is low also in both inefficient sub-groups. This shows that dairy farms are less vulnerable to policy changes than other sectors and have been integrated into the competitive market to a higher degree [3,16]. In total, the efficient farms achieve revenue of 4250 EUR per cow, 9.7% higher than the inefficient farms. The gross revenue for the whole farm sample (n = 47) is on average 4004 EUR/cow, which is in line with the findings of Poczta et al. [5], who reported gross revenue of 3870 EUR/cow for a typological group of large intensive farms that achieve similar milk yields.

**Table 4.** Composition of gross revenue.

| Gross Revenue (EUR/cow) | Efficient Farm (TE = 1.000) | Inefficient Farm (TE = 0.803) | Sub-Groups of Inefficient Farms | |
|---|---|---|---|---|
| | | | (TE = 0.881) | TE = 0.695 |
| Milk | 3709 (87.3% *) | 3366 (86.8%) | 3679 (87.7%) | 2862 (85.0%) |
| Calves and beef meat | 66 (1.6%) | 62 (1.6%) | 64 (1.5%) | 59 (1.7%) |
| Cows and veal meat | 286 (6.7%) | 230 (5.9%) | 241 (5.7%) | 213 (6.3%) |
| Subsidies | 189 (4.4%) | 217 (5.6%) | 207 (4.9%) | 234 (7.0%) |
| **Total (EUR/cow)** | **4250 (100%)** | **3875 (100%)** | **4191 (100%)** | **3368 (100%)** |
| **Total (EUR per farm)** | **610,237** | **551,388** | **630,072** | **436,902** |

* Numbers in parentheses indicate the percentage contribution in total gross revenue.

The cost structure and milk production cost for both efficiency groups and inefficient sub-groups are presented in Table 5. The share of capital cost in total expenses is 89% for both efficiency groups, but also for the sub-groups, confirming that modern dairy cow farms are capital intensive. The share of land rent and labor cost is the same in the efficient and inefficient farms, although the expenses per cow in value terms are higher for the inefficient farms (15% and 16.3% higher expenses for rent and wages, respectively). The analytical results indicate that feeding cost (expenses for purchased and home-grown feedstuff) is higher in the inefficient farms by 259 EUR/cow (EUR 1908 and EUR 2167 for the efficient and inefficient farms, respectively), although the latter achieve lower milk yield. This result indicates irrational and wasteful use of feedstuff as well as keeping animals of lower productivity in the inefficient farms. An interesting finding is that although the efficient farms cultivate more hectares for feed production, the inefficient farms spent more on home-grown feed, but also on purchased feed, which reveals the ineffective feeding management in the inefficient farms.

**Table 5.** Cost structure.

| Expenses Per Cow in EUR | Efficient Farm | Inefficient Farm | Sub-Groups of Inefficient Farms | |
|---|---|---|---|---|
| | | | TE: 0.80–1.00 (n = 18) | TE < 0.80 (n = 13) |
| I. Land rent | 107 (3.2% *) | 123 (3.1%) | 111 (2.8%) | 142 (3.5%) |
| II. Labor wages | 264 (7.8%) | 307 (7.8%) | 319 (8.1%) | 289 (7.2%) |
| III. Purchased feed | 1628 (48.2%) | 1827 (46.0%) | 1811 (46.0%) | 1851 (46.0%) |
| IV. Variable capital | 615 (18.2%) | 815 (20.5%) | 825 (20.9%) | 799 (19.9%) |
| *Home-grown feed* | *280* | *340* | *331* | *354* |
| *Other expenses* | *335* | *475* | *494* | *445* |
| V. Fixed capital | 765 (22.6%) | 895 (22.6%) | 868 (22.1%) | 938 (23.3%) |
| Total Expenses ow) | 3381 (100%) | 3967 (100%) | 3934 (100%) | 4019 (100%) |
| Milk cost (EUR/lt) | 0.346 | 0.449 | 0.415 | 0.514 |

* Numbers in parentheses indicate the percentage share of each kind of expenses to total cost.

Another interesting finding is that the fixed cost per cow, which is the second most important cost driver after feeding cost, is much higher for inefficient farms (especially for farms in the second inefficient sub-group), although total fixed costs are allocated to the same herd size between the two groups. Although their share over total expenses does not differentiate between the efficiency groups, it indicates either excessive investments on fixed capital (mainly buildings and machinery) for inefficient farms, which calls for herd size expansion, and/or that the break-even point for these investments will occur in the future. Sauer and Latacz-Lohmann [47] and Frick and Sauer [50] have shown in recent years that dairy cattle farmers are not risk averse in their investment decisions, showing an increasing willingness to invest in new equipment and techniques, increasing their fixed cost. In general, our results show that efficient farms are better organized, as they spend on average EUR 3381 annually for their operation, 14.8% less than the inefficient farms. The total expenses for the whole farm sample were on average 3767 EUR/cow, a result in line with that of Poczta et al. [5], who reported total expenses of 3833 EUR/cow for dairy farms of similar production systems. In addition, the lowest milk production cost, calculated with the adoption of the proportional costing method, occurred in the case of the efficient farms: EUR 0.346 for the production of one liter of milk, 23% less than the inefficient farms. The latest study on the cost of milk production in eight key milk-producing centers in the EU shows that in 2019, milk production cost was between 0.342 and 0.586 EUR/lt, indicating a large variation among countries mainly due to different farming structures [9]. The average milk production cost in the EU was 0.453 EUR/lt, a finding similar to ours for the inefficient farms, whereas the average farm-gate milk price in the EU was 0.358 EUR/lt, indicating a large cost shortfall (23.8%).

The financial results are summarized in Table 6. As already mentioned, the highest gross revenue was achieved by the efficient farms, indicating that a higher level of efficiency is related to a higher value of production. This is in accordance with the findings of Theodoridis and Ragkos [23], Siafakas et al. [24] and Mitsopoulos et al. [25], who estimated efficiency in modern Greek dairy cattle farms. Consequently, gross margin (gross revenue less than the variable cost) was increased in the efficient farms by 63%, from 1233 to 2007 EUR per cow, indicating the importance of management skills in running a dairy farm, especially in times of crisis where soaring inflation squeezes budgets of farmers and rises in the cost of fuel, feedstuff and agrochemical inputs outstrip any increase in milk price.

**Table 6.** Financial results.

| Financial Results (EUR/Cow) | Efficient | Inefficient | Sub-Categories of Inefficient Farms | |
|---|---|---|---|---|
| | | | TE: 0.80–1.00 (n = 18) | TE < 0.80 (n = 13) |
| Gross revenue | 4250 | 3875 | 4191 | 3368 |
| Variable cost | 2243 | 2642 | 2636 | 2650 |
| Gross margin | 2007 | 1233 | 1555 | 718 |
| Fixed cost | 1136 | 1325 | 1299 | 1369 |
| Profit or loss | 871 | −92 | 256 | -651 |

The net economic margin (profit or loss) shows that only the farms that efficiently utilize the existing technology will be economically viable in the long term, exhibiting on average a profit of 871 EUR/cow compared to inefficient farms that exhibit on average a loss of 92 EUR/cow. However, as expected, the analytical results for the two inefficient sub-groups show that the very inefficient farms exhibit a great loss of 651 EUR/cow, while the less inefficient farms (group with TE from 0.80 to 0.99) show a profit of 256 EUR/cow. These farms, although sustainable in the long run, must enhance their managerial abilities and adopt best-observed practices to improve their competitiveness.

## 4. Conclusions

This study identified the most efficient dairy farms and described their main structural and economic characteristics. The appropriate farm structure, the major cost drivers and the main features of a profitable dairy farm were revealed. The efficiency analysis was conducted using farm accounting data from 47 dairy farms in Greece and confirmed that there is still room for improvement in the intensive dairy production system in the EU. The analysis showed that dairy cattle farms should operate with increased variable cost and utilize their infrastructure at full capacity to decrease their fixed cost per cow. Farms should diversify their products to increase their economic resilience and be less risk averse in an environment where margins to lower production costs and to improve competitiveness in highly intensive farms have been narrowed down. The results verify that the dairy cattle farms of entrepreneurial mindset have high growth potential. Successful management of a modern dairy cattle farm requires the design of integrated strategic plans, lifelong training of farmers, close consultation with the processing industry and extensive services, the adoption of new technological trends such as digital technologies, Internet of Things and decision support tools and the implementation of best practices and innovations. Dairy cattle farmers must continue undertaking investments and become more innovative and market oriented. Finally, the proper functioning of an efficient supply chain in dairy products, deprived of unfair practices, is necessary for a resilient and sustainable dairy cattle sector.

Cow milk production in the EU is expected to continue increasing, and it will reach 162 million tons by 2030 (+0.6% per year), contributing to further consolidation of the sector [51]. Nevertheless, the EU dairy cattle sector is facing severe challenges related to public policies, environmental restrictions, changing consumer demands, input and output price volatility and increased competition. The European Green Deal in the form of the Farm to Fork Strategy calls on farms to ensure quicker implementation of the changes required to achieve climate-neutral, environment- and resource-friendly livestock farming. To address these new challenges, dairy cattle farmers must adopt and implement best-observed and innovative practices that will improve animal welfare, promote biodiversity and protect the environment, but at the same time secure a fair income and increase profitability. In addition, dairy farmers should revise their competitiveness under new perspectives and beyond price, emphasizing the promotion of sustainability and the production of high value-added and "greener" products, adapting to consumer demand. Towards this direction efficiency analysis, through the identification of the best farms and the practices that these

farms apply can be a very practical tool. Hence, an expansion of the EU product quality schemes and geographical indications (European Commission, 2016r) might represent an opportunity for EU dairy farming. The spread of dairy farming across the entire EU might furthermore open opportunities for using this farm type in a more pronounced fashion for landscape management and preservation.

**Author Contributions:** Conceptualization, G.K., G.A. and A.T.; Data collection and processing, G.K. and A.R., Data analysis, G.K. and A.T.; Writing of first draft G.K. and A.T.; Writing—original draft preparation, G.K., A.R. and A.T.; All authors have read and agreed to the published version of the manuscript.

**Funding:** This research has been co-financed by the European Union and Greek national funds through the Operational Program Competitiveness, Entrepreneurship and Innovation, under the call RESEARCH—CREATE—INNOVATE (project acronym RawCheese; project code: T1EDK-03989).

**Data Availability Statement:** The data presented in this study are available on request from the corresponding author. The data are not publicly available as they were collected by G.K. for her Ph.D. thesis with no external funding and will be used for further processing.

**Conflicts of Interest:** The authors declare no conflict of interest.

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
