# Peer review of "Entrepreneurship in Dairy Cattle Sector: Key Features of Successful Administration and Management"

_land, doi:10.3390/land11101736_

Round 1

Reviewer 1 Report

This is an interesting paper that analyzes detailed technical and economic data at the farm level concerning one of Greek agriculture's most important sectors, dairy farming. The importance of such an analysis is also underlined by the lack of similar data in established databases such as FADN: within the timespan 2004-2020, only four years of data are reported for the Farming Type "Specialist Milk" in Greece. An additional strength of the paper is the exposure of divergent views (and data) on the issue of own feed production on a dairy farm.

Still, some issues need to be addressed by the authors to make this work more convincing. These issues and the proposed clarifications and additions are as follows:

1.       The authors rely their analysis on an output-oriented measure of technical efficiency. However, given the VRS assumed, an input-oriented DEA would produce different technical efficiency scores. Did the authors estimate the model using an input-oriented DEA, as well? If yes, are there any significant differences in the results? Overall, what is the reasoning behind author’s decision to adopt an output-oriented measure rather than an input-oriented measure of technical efficiency?

2.       The authors rely on the estimation of technical efficiency scores to assess the economic performance and competitiveness of Greek dairy farms. However, it is quite common in studies of efficiency and productivity to analyse technical efficiency within a broader productivity decomposition framework since productivity change is perceived as a more complete measure of economic performance accounting not only for efficiency changes but also for economies of scale and technological change. A short discussion justifying the author’s choice to focus on efficiency rather than productivity growth would be a useful addition.

3.       In the introductory section, the authors state: “…the efficient use and the rational allocation of the existing resources in dairy farms is also crucial to their competitiveness.” However, in the empirical analysis, the authors focus only on the technical inefficiency of dairy farms neglecting to account for the possible presence of allocative inefficiencies that might be of equal importance.

4.       I have some serious concerns about the choice and construction of output and inputs used in the implementation of the DEA approach, that, needless to say, may have a significant impact on the results of the study. Focusing first on output, the authors use gross revenues as a proxy of the output variable. Although it is not clear in the text how exactly this variable has been constructed, it is my understanding that gross revenue variable is a measure of the short-run profits of the farm. However, the purpose of the analysis is to approximate the production frontier (production technology) and next benchmark farms relative to this production frontier. Within this context, the use of gross revenues as the output variable seems to be improper. If dairy farms produce more than one outputs, then a multi-output DEA model should be employed using produced quantities as outputs. Alternatively, an aggregated output can be constructed using revenue shares as weights in the aggregation procedure. A third option would be to rely on value added measures.

 5.      Moreover, variable and fixed costs are used as inputs in the analysis. However, this aggregation scheme is based on very strong assumptions and at the same time sacrifices a lot of information hidden in the dataset which may have a significant effect on the results of the study. I would strongly recommend the authors to introduce different categories of inputs in their analysis and rely on Divisia index methods to aggregate quantities of inputs, if necessary.

6.       The authors use human labor as an input defined as hours per year. Does this include both hired and family labor? If yes, are those two inputs perfect substitutes? If not, then it would be improper to aggregate them. If hired labor is included, why isn’t it included in the variable cost variable?

7.       The authors use the DEA approach to obtain estimates on technical efficiency. However, this approach does not control for stochastic effects and measurements errors that are likely to be large in this study (see my earlier comments on inputs aggregation). Probably, a better option would be to use a stochastic frontier approach (SFA) which accommodates additionally a random variable in the estimation of the production technology treating thus deviations from the frontier as comprising both random error (white noise) and inefficiency. In any case, a brief discussion explaining your decision to rely on a DEA approach would be useful.

8.       The authors use the efficiency estimates obtained from the application of DEA approach and then use these estimates to classify farms into two groups (i.e., fully efficient and inefficient group). Next, they calculate performance indicators and other variables for each group and compare them to reveal the characteristics of the most successful (efficient) dairy farms. However, since the interest is on farmers’ characteristics that affect efficiency scores, it would be more reasonable to run a second stage tobit regression using efficiency scores obtained from DEA as dependent variable and next calculate the marginal effects of farmers’ characteristics on efficiency. This seems a more reasonable option since the current classification seems somehow arbitrary. For instance, why using only two groups and not more than two groups of farmers? In this setting, a farmer with an efficiency score 99% is considered as an inefficient farmer that is the same with a farmer presenting a 10% efficiency score. In other words, transforming a continuous variable into a binary variable results in loosing important information and I do not see any reason to do so.

9.       Please provide more information on the sampling methods used. Did you rely on random sampling techniques? Is self-selection an issue here?

Author Response

-

Reviewer 2 Report

This paper can be a management guide for farmers. The approach is a good one, healthy from a managerial point of view.

Author Response

-

Reviewer 3 Report

The article has some deficiencies in terms of its structure, the most important being the lack of a discussion section. If the results and discussion sections are together, the section's title should be changed. On the other hand, the data has not been sufficiently exploited to obtain information that shows to a greater extent, the variables responsible for the efficiency of the farms. Finally, the authors must justify the sample's selection and size.

Line 98. It would be interesting to include more information about the survey. Indicate why a sample of 47 farms was taken and whether their selection was random or for convenience.

Line 115. Why was a model oriented towards output and not input chosen?, justify.

Line 127: Use subscripts

Line 177-204. Much of these paragraphs correspond to the discussion.

Table 3. Include dispersion measure.

Line 219. Although dividing the farms into two groups is correct, proposing a division into three could show some other tendencies. It would be interesting to propose some correlations between efficiency and input and output variables.

Line 291. Include some statistical tests to validate this statement

Line 321. Why were parameters associated with cow fertility not evaluated? It would be interesting to know if this is associated with lower productivity per cow.

Line 151. The results section includes the discussion; therefore, it is advisable to separate or change the section's title.

It would be interesting to go deeper into the reasons that explain the inefficiency of the farms, perhaps categorizing them by different variables, for example, size, milk production per cow, production per hectare, etc.

Author Response

-

Reviewer 4 Report

The paper should be published with minor changes.

The article is well-written, clear, and gives robust results.

The paper uses data collected through a farm management survey in Greece. Data Envelopment Analysis is applied on dairy cattle farms to estimate their efficiency in utilizing the available resources and to identify the most efficient ones. A comparative financial analysis is used between the efficient and inefficient farms highlighting the appropriate farm structure and determining the major cost drivers in modern dairy cattle farming.

The scientific content is accurate, balanced, and attractive. The description of the method is used sufficiently. The quality of writing is good, with clearness. The significant value of this work and the importance of its results are essential for developing the strategy for the dairy farming sector.

The paper needs some minor changes. First, the introduction should include information about the Greek dairy cattle sector. The conclusions section should be improved to give the practical solutions you promised and clearly to have the critical features of successful administration and management. Finally, you have to provide policy recommendations for a resilient and profitable dairy sector.

Author Response

-

Reviewer 5 Report

1. Add subtitles to increase readability.

2. why there is no discussion section?

3. Why the authors use data envelopment analysis to measure efficiency, and why not consider stochastic frontier analysis?

4. What are the limitations of the paper?

Author Response

-

Round 2

Reviewer 1 Report

The revised paper provides sufficient revisions/additions as regards my previous comments and suggestions, No. 1, 2, 7, and 9. As far as comment No 3 is concerned, the authors say that the allocation of resources has nothing to do with allocative efficiency. Obviously, it has something to do with it. I would expect them to say that the paper's focus is on technical efficiency only and that allocative efficiency is beyond the scope of this study and to remove this phrase so as not to cause confusion. But this is minor, so let's say OK.

However, there are still some unresolved issues with four of my previous comments:

Comment 4: The authors use the revenues as output, in which the subsidies are included. First, if you use revenues, you must run an equivalent of the revenue function instead of the production function, which means that you must have the output values in your independent variables. If, for example, two farms in the sample use the same amounts of inputs and produce the same amounts of outputs, but sell the products at different prices, then one will be efficient and the other inefficient, which is incorrect. It's even worse with subsidies in it. In the same example, if one receives subsidies and the other does not, then the one receiving it will turn out to be efficient and the other not, which is at least misleading, as both produce the same quantities of products. That is why I initially suggested that the authors use a multi-output DEA with the output quantities or at least sum all the quantities into one, using the revenue shares.

Comment 5: The same (as the above argument) goes for inputs. It is wrong to put variable cost and fixed cost as inputs. This should go without saying. The important thing in all of this is that the results would have been entirely different if the authors had made the necessary corrections.

Comment 6: The authors probably didn't understand what I said. The authors use the number of animals, labor, variable costs and fixed costs as inputs. Family labor is considered a quasi-fixed input, and hired labor is considered a variable input in the literature. So why didn't these go into their last two inputs that they call variable and fixed costs and go in separately? That's what I asked. Of course, the correct action is to enter them separately, as it is now, and omit the last two inputs that they call variable and fixed costs, as they are not inputs. Instead of these, the intermediate inputs should be entered cumulatively using indicators, and capital.

Comment 8: I don't see why the authors can't make the estimate I suggested. Also, their choice to divide them into efficient and inefficient in the way they do remains arbitrary. It makes no sense to say that someone who is 99% effective is in the same group as someone who is 10% effective and find their characteristics as if those two are the same. The authors keep saying about the small sample as excuses, but this is a drawback of the study, not an excuse not to make corrections. 

Reviewer 3 Report

The authors have adequately responded to my comments, so I suggest approving the manuscript in its current form.

Reviewer 5 Report

Section “3. Results and Discussion”  should add subtitle, it's hard to distinguish between results and discussions.
